# Prediction of COVID-19 spreading profiles in South Korea, Italy and Iran by data-driven coding

Choujun Zhan[1], Chi K. Tse[2¤]*, Zhikang Lai[3], Tianyong Hao[1], Jingjing Su[4]

**1** School of Computing, South China Normal University, Guangzhou, China, **2** Department of Electrical Engineering, City University of Hong Kong, Hong Kong, China, **3** School of Electrical and Computer Engineering, Nanfang College of Sun Yat-Sen University, Guangzhou, China, **4** Nethersole School of Nursing, The Chinese University of Hong Kong, Hong Kong, China

¤ Current address: Department of Electrical Engineering, City University of Hong Kong, Kowloon, Hong Kong, China

* chitse@cityu.edu.hk

**Data Availability Statement:** The data underlying the results presented in the study are available from https://www.who.int/emergencies/diseases/novel-coronavirus-2019.

## Abstract

This work applies a data-driven coding method for prediction of the COVID-19 spreading profile in any given population that shows an initial phase of epidemic progression. Based on the historical data collected for COVID-19 spreading in 367 cities in China and the set of parameters of the augmented Susceptible-Exposed-Infected-Removed (SEIR) model obtained for each city, a set of profile codes representing a variety of transmission mechanisms and contact topologies is formed. By comparing the data of an early outbreak of a given population with the complete set of historical profiles, the best fit profiles are selected and the corresponding sets of profile codes are used for prediction of the future progression of the epidemic in that population. Application of the method to the data collected for South Korea, Italy and Iran shows that peaks of infection cases are expected to occur before mid April, the end of March and the end of May 2020, and that the percentage of population infected in each city or region will be less than 0.01%, 0.5% and 0.5%, for South Korea, Italy and Iran, respectively.

## 1 Introduction

The 2019 Coronavirus Disease (COVID-19) is a highly contagious disease caused by infection of the SARS-CoV-2 virus. The disease began to spread in China in mid December 2019 [1], and as the volume of intercity travel escalated around the Lunar New Year period, the number of infected individuals began to soar in mid January 2020 [2, 3] With no travel restriction in place due to the low level of vigilance or unawareness of the disease during the early phase of the outbreak, the spreading of the disease had gone almost unobstructed. Travel restriction began to be implemented throughout China since January 24, 2020, which has proven to be effective in curbing the spread of the virus. However, international traffic has not ceased and infectious individuals (who may or may not show any symptom at the time of travel) have

**Funding:** CZ was supported by National Science Foundation of China Project 61703355 (http://www.nsfc.gov.cn/) and [Science and Technology Program of Guangdong 201904010224.] CKT was supported by City University of Hong Kong under Special Fund 9380114 (https://www.cityu.edu.hk/). The funders had no role in study design, data collection and analysis, decision to publish, or preparation of the manuscript.

**Competing interests:** The authors have declared that no competing interests exist.

actually travelled to different countries with the virus they contracted [4, 5]. Recent studies have also showed that travel restriction did contribute to the control of the spreading of COVID-19 within China as well as in a global context [4, 6, 7]. By May 17, 2020, China had confirmed a total of 82,954 cases of COVID-19 infection, with death toll reaching 4,634. While China has begun to see declining numbers of infected cases in most cities from late February, other countries started to report surging number of cases in some cities or regions. As of May 17, 2020, the cumulative number of cases of COVID-19 infection was 4,804,011 worldwide, with South Korea, Italy and Iran reporting surges of infected cases within two weeks in late February and early March. Moreover, the global mortality rate has increased from around 3% (March 8, 2020) to 6.84% (May 14, 2020).

In our recent work (available on February 19, 2020) [3], intercity travel data obtained from Baidu Migration [8] has been collected and integrated into the traditional Susceptible-Exposed-Infected-Removed (SEIR) model [9–11] to account for the effects of inflow and out-flow traffic between 367 cities in China. Parameters of susceptible-to-exposed infection rates, exposed-to-infected infection rates, and recovery rates for 367 cities have been identified by fitting the augmented SEIR model with historical data available from the National Health Commission of China. The predicted spreading profiles of the 367 Chinese cities (which were available on February 19, 2020 [3]) have been highly consistent with the actual profiles, including the times of infection peaks and the percentages of infected individuals in the 367 cities.

In this work, we build on the data and estimated parameters obtained for 367 cities in our prior work [3], and establish a library of profiles (sets of codes) of different spreading profiles. Suppose a new outbreak has occurred in a given population. The numbers of infected and recovered cases during the early phase of the spreading form an initial profile. This initial or incomplete profile is compared with the historical full profiles obtained previously. Then, by selecting the best fit historical profiles, we identify the candidates parameters for prediction of the future spreading profile in that given population (of a city or region). It should be emphasized that the set of historical profiles obtained previously covers various possible spreading dynamics, representing a variety of contact topologies and transmission mechanisms including the travel effects that have been integrated in the model used to capture the transmission dynamics of the 367 cities. Thus, a new outbreak in a given population would likely follow one or a combination of the profiles based on the augmented model proposed in our previous work [3], and hence can be reconstructed from the historical sets of profiles. In this work, we develop a procedure for implementing the selection of historical profiles, identification of best-fit parameters and construction of future profile. We have applied the procedure to predict the epidemic progression in the cities of South Korea, Italy and Iran. Results of this study showed that the first wave of epidemic progression in most cities in South Korea peaked between early March and early April, 2020, and in Italy between late April and mid May, 2020, while Iran would have its peak in late April, 2020. We have also investigated the number of infected individuals in each city or region. Our method provides the average number of individuals eventually infected, along with a predicted deviation range at 95% confidence level. For Korea, we predicted that Daegu and Gyeongsang North Road would have around 7,619 and 1,287 people eventually infected (i.e., 0.306% and 0.062% of the city's population), respectively, whereas the number of infected individuals in other cities in South Korea would be fewer than 300, i.e., less than 0.01% of city population. For Italy, we predicted that Lombardy and Emilia-Romagna would eventually have about 90,000 and 30,000 infected cases (i.e., 0.802% and 0.604% of the region's population), respectively, and the number of people eventually infected in other cities in Italy would be below 10,000. Moreover, Iran would have more than 9,000 and 6,000 confirmed cases in Tehran and Isfahan (i.e., 0.39% and 0.2% of the city's population). In addition, the number of people infected in most other cities would be larger than 1,000

($>$0.1% of the city's population). From the progression trends of the epidemic in these three countries, provided control measures continue to be in place, our model show that the number of people infected in most regions of these three countries will peak before the end of May 2020. Hence, the first wave of the spreading of the disease is expected to come under control before the end of May 2020.

In the remainder of the paper, we first introduce the official daily infection data used in this study. The augmented SEIR model is briefly reviewed, mainly to introduce the parameters of the model used for prediction of spreading profiles. The key procedure for matching historical profiles and prediction of future spreading profiles will be explained. Results of application of the proposed method to prediction of the peaks and extents of outbreaks in South Korea, Italy and Iran will be given. Finally, we will provide a discussion of our estimation of the propagation and the reasonableness of our estimation in view of the measures taken by the authorities in controlling the spreading of this new disease.

## 2 Data

The World Health Organization currently sets the alert level of COVID-19 to the highest, and has made data related to the epidemic available to the public in a series of situation reports as well as other formats [12]. Our data include the number of infected cases, the cumulative number of infected cases, the number of recovered cases, and death tolls, for individual cities and regions in South Korea and Italy, from February 19, 2020, to May 12, 2020, and in Iran from February 19, 2020, to March 22, 2020, Data organized in convenient formats are also available elsewhere [13–15]. Samples of data for Daegu, Gyeongsang North Road (South Korea), Lombardy, Amelia Romagna, Tehran and Mazandaran are shown graphically in Fig 1. It should be noted that the data obtained for South Korea, Italy and Iran correspond to initial stages of the epidemic progression as the number of infected cases are still climbing, as of March 6, 2020.

At present, there are two types of tests for confirming COVID-19 infected cases. One type of tests aims to confirm the presence of the SARS-Cov-2 virus in the body of an individual, which is commonly done via detecting the viral RNA through a polymerase chain reaction (PCR) [16]. The other type establishes the presence of antibodies in an individual, i.e., whether the individual being tested has been infected in the past, regardless of him or her carrying the virus at the time of testing. In this work, the official number of infected cases corresponds only to individuals who have been tested positive for the presence of the SARS-Cov-2 virus.

## Method

### The augmented SEIR model

The travel-data augmented SEIR model [3] describes the spreading dynamics in terms of a basic fourth-order dynamical system with consideration of intercity travel in China. Consider a city $j$ of population $P_j$. The states of the model are the number of susceptible individuals $I_j(t)$, the number of exposed individuals (infectious but without symptom) $E_j(t)$, the number of infected individuals $I_j(t)$, and the number of recovered or removed individuals $R_j(t)$. The model takes the following form in discrete time [3, 17]:

$$X_j(t+1) = F_{\mathrm{aSEIR}}(X_j(t), M_j, \mu_j) \tag{1}$$

where $X_j(t) = [S_j(t)E_j(t)I_j(t)R_j(t)]^T$ is the state vector on day $t$, $F_{\mathrm{aSEIR}}(.)$ is the travel-data augmented function, $M_j$ is the set of inflow and outflow travel strengths for city $j$, and $\mu_j$ is the set of parameters for city $j$, i.e.,

$$\mu_j = [\alpha_j, \beta_j, \kappa_j, \gamma_j, \delta_j, k_l] \tag{2}$$

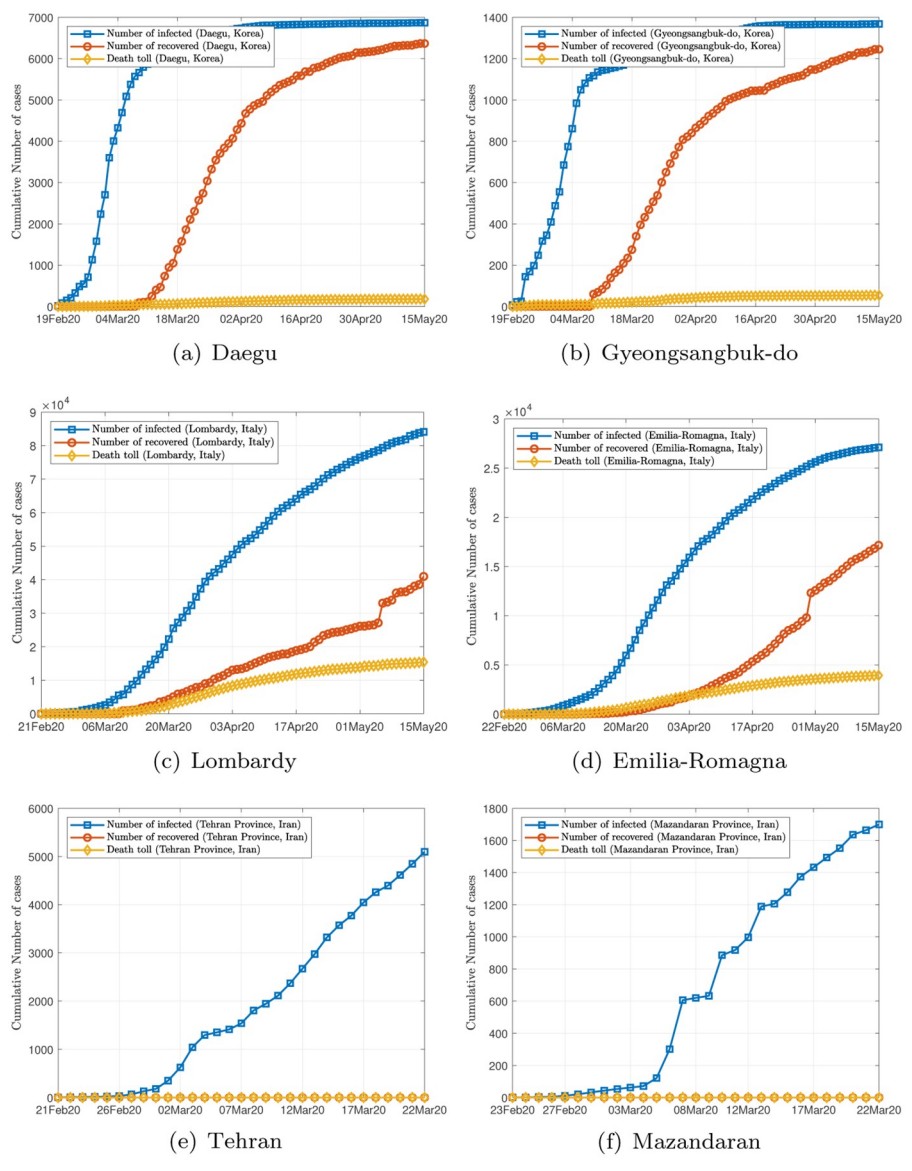

**Fig 1. Samples of data.**

where $\beta_j$ is the rate at which a susceptible individual is infected by an infected individual in city $j$, $\alpha_j$ is the rate at which a susceptible individual is infected by an exposed individuals in city $j$, $\kappa_j$ is the rate at which an exposed individual becomes infected in city $j$, and $\gamma_j$ is the recovery rate in city $j$, $k_I$ is the possibility of an infected individual moving from one city to another, and $\delta_j$ is the eventual percentage of the population infected in city $j$. Moreover, the eventual infected population in city $j$ is given by $N_j^s = \delta_j P_j$.

To facilitate comparison and matching of profiles, we introduce the normalized states as $\frac{\bar{I}_j(t)\Delta}{=I_j(t)/N_j^s(t)}$, $\frac{\bar{E}_j(t)\Delta}{=E_j(t)/N_j^s(t)}$, $\frac{\bar{S}_j(t)\Delta}{=S_j(t)/N_j^s(t)}$, and $\frac{\bar{R}_j(t)\Delta}{=R_j(t)/N_j^s(t)}$. Thus, (1) can be represented in *normalized form* as

$$\bar{X}_j(t+1) = \mathcal{F}_{\mathrm{aSEIR}}(\bar{X}_j(t), M_j, \mu_j) \qquad (3)$$

where $0 \leq |\bar{X}_j(t)| \leq 1$. Since the above model has taken into account the human migration

effect as well as the necessary transmission mechanism, we may consider the basic set of parameters to represent the characteristics of the propagation profile of city $j$. The complete set of parameters have been identified for 367 cities in China [3], which will serve as a set of codes for various propagation profiles of COVID-19 so far obtained. For brevity, we do not repeat the results here.

While two different cities may have different population size and percentage of eventual infected population, the rates of infection and recovery should be similar across a group of cities, i.e., $\mu_i \approx \mu_j$. Thus, in normalized form, we have

$$\| \bar{X}_i(t) - \bar{X}_j(t) \| < \epsilon \ \text{ for some } \epsilon > 0, \tag{4}$$

for cities $i$ and $j$ within a group of cities having similar parameter sets. This also means

$$N_j^s X_i(t) \approx N_i^s X_j(t) \ \text{ or } \ \delta_j P_j X_i(t) \approx \delta_i P_i X_j(t) \tag{5}$$

for the group of cities having similar rates of infection and recovery. Thus, provided the historical archive has adequately covered the possible dynamical profiles, we are able to perform fast prediction for any city $o$, by fitting an incomplete set of data (corresponding to an early outbreak stage in city $o$) and using the model parameters already obtained previously, as detailed in the following subsection.

## Prediction method

The proposed data-driven prediction algorithm is based on the set of historical data of the spreading profiles of COVID-19 in 367 cities in China, namely, 367 sets of normalized time series of the form:

$$
\begin{aligned}
\bar{I}_i^{(c)} &= \{\bar{I}_i(1), \bar{I}_i(2), \ldots \bar{I}_i(K_i)\} \\
\bar{R}_i^{(c)} &= \{\bar{R}_i(1), \bar{R}_i(2), \ldots \bar{R}_i(K_i)\}
\end{aligned}
\tag{6}
$$

where $i = 1, 2, \ldots, 367$, and $K_i$ is the length of the data recorded in city $i$. Superscript "$(c)$" denotes data of Chinese cities.

Now, suppose an outbreak occurs in city $o$, and only $k_o$ days of data have been obtained in normalized form as

$$
\begin{aligned}
\bar{I}^{(o)} &= \{\bar{I}_o(1), \bar{I}_o(2), \ldots \bar{I}_o(k_o)\} \\
\bar{R}^{(o)} &= \{\bar{R}_o(1), \bar{R}_o(2), \ldots \bar{R}_o(k_o)\}
\end{aligned}
\tag{7}
$$

where $k_o < K_i$. Then, assuming the spreading profile of city $o$ is related to that of city $i$ in the historical archive, as permitted by virtue of the validity of (4), we formulate the following optimization problem to predict the epidemic progression in city $o$:

$$
\begin{aligned}
\mathrm{P}_0: \quad &\min_{i \in (1,N)} f_i \\
&\text{s.t.} \quad \text{(i)} \ f_i = \sum \left[ w_I (\bar{I}_o(j) - \bar{I}_i^{(c)}(j))^2 + w_R (\bar{R}_o(j) - \bar{R}_i^{(c)}(j))^2 \right] \\
&\qquad\quad \text{(ii)} \ \mu_L \leq \mu \leq \mu_U \\
&\qquad\quad \text{(iii)} \ w_I, w_R > 0.
\end{aligned}
\tag{8}
$$

where $N$ is the number of Chinese cities in the historical archive, $w_I$ and $w_J$ are weighting coefficients, $\mu_L$ and $\mu_U$ are the lower and upper bounds of the searching space, respectively. By

solving the the nonlinear optimization problem, we can find the most closely resembling growth curve from the historical profiles, e.g., city $i$. Then, we apply the the augmented SEIR model with the profile code given in the parameter set for city $i$ to predict the future spreading trend of city $o$. Furthermore, we can choose the top $n$ best candidates with the smallest error as the candidate set for prediction, giving an average predicted propagation profile and a deviation range based on $n$ best-fit profile codes.

## 4 Results

We apply the aforedescribed procedure to the data obtained so far for cities or regions in South Korea, Italy and Iran, as listed in Table 1. For each city or region, we identify a group of 10 profiles of best fit from the historical archive, and retrieve the corresponding sets of profile codes for generating the propagation profiles in the coming days. Using these 10 profiles, we produce an average progression profile, which is also accompanied by a deviation range at 95% confidence level. For Iran, we have only collected data for cities and provinces up to March 22, 2020, as data after this data was no longer available for individual cities and provinces.

Figs 2, 3 and 4 show the data and the predicted number of infected individuals, each with a deviation range of the predicted average trajectory, for South Korea, Italy and Iran, respectively, and Figs 5, 6 and 7 show the corresponding cumulative values. Statistics of infection peaks are shown in Fig 8. Statistics of the percentage of population eventually infected and the number of individuals eventually infected are shown in Fig 9. Our key findings are summarized as follows:

1. The number of active infected individuals in South Korea, Italy and Iran is expected to continue to increase until the end of May 2020. While South Korea saw peaks in most of the cities between March 8, 2020 and April 9, 2020, most cities in Italy saw peaks between April 15 and mid May, while most provinces in Iran would saw peaks before the end of May 2020, as shown in the distributions of the peak times given in Fig 9(a) to 9(c).

2. For South Korea, our results show that Daegu (with a population of 2,487,823) and Seoul (with a population of 10,018,537) are the two hardest hit cities, with 7,619 ±2,096 and 1,287 ±197 people eventually infected, accounting for 0.306% ±0.084% and 0.062% ±0.009% of the city's population. In other cities in South Korea, the number of infected people will be fewer than 300, i.e., below 0.01% of the population.

3. For Italy, our results show that Lombardy (with a population of 10,078,012) and Emilia-Romagna (with a population of 4,459,477) have the highest number of cases, with about 90,000 and 30,000 people eventually infected, accounting for 0.802% and 0.604% of the region's population. In other Italian cities or regions, the number of infected people will be below 10,000.

4. For Iran, we expect Tehran (with a population of 13,267,637) and Isfahan Province (1,292,283) to be most severely affected, reaching more than 9,000 (0.067%) and 1,695 ±92 (0.13%) cases eventually, respectively. Other Iranian cities will see more than 1,000 eventual infected cases. Our prediction shows that about 0.5% of the country's population have been infected until May 14, 2020.

5. Provided the authorities continue to impose strict control measures, our model shows that most regions (cities and provinces) of these three countries would see peaks of infection cases before the end of May 2020. Hence, the first wave of epidemic will come under control by June 2020 for these three countries.

**Table 1. Populations of cities, regions or provinces in South Korea, Italy, and Iran.**

| City/Region/Province | Population | City/Region/Province | Population |
|---|---|---|---|
| *South Korea:* | | *Italy:* | |
| Daegu | 2,487,823 | Lombardy | 10,078,012 |
| Seoul | 10,018,537 | Venetia | 4,905,854 |
| Gwangju | 1,472,802 | Emilia-Romagna | 4,459,477 |
| Busan | 3,513,361 | Piedmont 4,356,406 | |
| Gyeongsangbuk-do | 2,071,424 | Lazio | 5,879,082 |
| Gyeongsangnam-do | 2,870,401 | Tuscany | 3,729,641 |
| Chungcheongbuk-do | 1,191,341 | Sicily | 5,029,675 |
| Chungcheongnam-do | 606,019 | Trento | 539,898 |
| Jeollanam-do | 1,055,957 | Liguria | 1,565,349 |
| Jeollabuk-do | 652,858 | Marche | 1,532,000 |
| Gangwon-do | 1,135,134 | Campania | 5,827,000 |
| Incheon | 2,927,295 | Abruzzo | 1,315,000 |
| Jejudo | 666,686 | Apulia | 4,048,000 |
| Gyeonggi-do | 12,476,073 | Umbria | 884,600 |
| Daejeon | 1,518,024 | Molise | 330,000 |
| Ulsan | 1,173,568 | Basilicata | 595,727 |
| Sejong | 314,126 | Friuli Venezia Giulia | 1,216,000 |
| | | Sardinia | 1,648,000 |
| *Iran (Provinces):* | | Calabria | 1,957,000 |
| Tehran | 13,267,637 | Aosta Valley | 126,200 |
| Mazandaran | 3,283,582 | Bolzano | 520,900 |
| Bushehr | 2,712,000 | | |
| Golestan | 1,868,819 | | |
| Semnan | 702,360 | | |
| Isfahan | 5,120,850 | | |
| Fars | 4,851,000 | | |
| Hormozgan | 1,776,000 | | |
| Bushehr | 1,163,400 | | |
| Gilan | 2,530,696 | | |
| Ardabil | 1,270,420 | | |
| Kurdistan | 1,603,000 | | |
| Markazi | 1,429,000 | | |
| Khuzestan | 4,711,000 | | |
| Lorestan | 1,754,000 | | |
| Razavi Khorasan | 5,994,000 | | |
| Sistan and Baluchestan | 2,775,000 | | |
| East Azerbaijan | 3,725,000 | | |
| West Azerbaijan | 3,081,000 | | |
| Kerman | 3,164,718 | | |
| Qom | 1,292,283 | | |

Our prediction on the South Korean cities has revealed a very rapid progression of the epidemic, with 5,000 infections emerged within 10 days and peaks to be expected in most cities or regions within about 2 weeks. The Korean authorities have managed to test an overwhelmingly large number of people (140,000 until March 5, 2020) within a short time, thus preventing a large number of infected and infectious individuals not being quarantined in time [18]. This

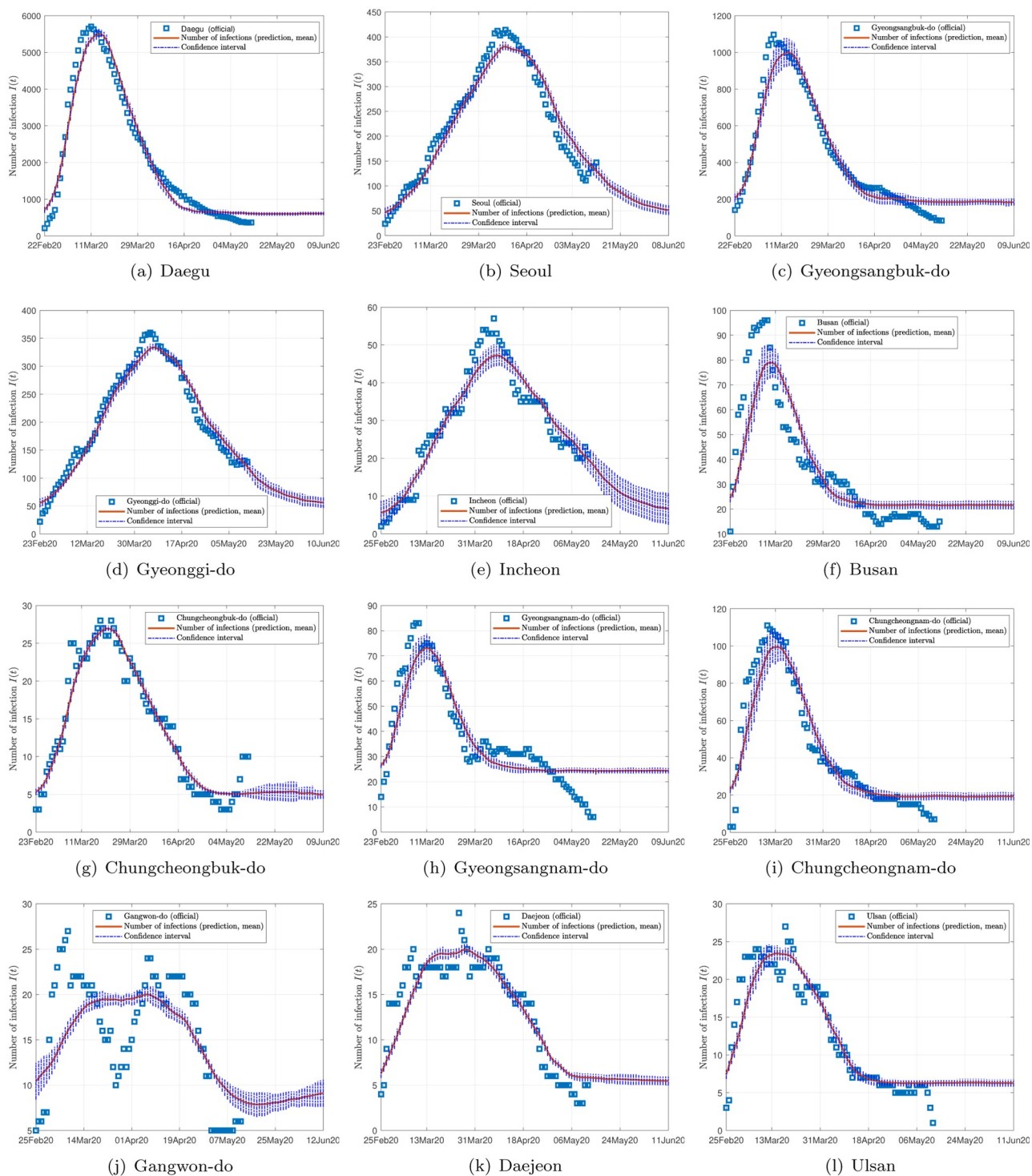

**Fig 2. Official and estimated number of infected individuals in some cities or regions in South Korea.**

strategy has an obvious advantage of offering a clear picture of the extents and locations of the infected individuals in the country at the early phase of the epidemic progression. The epidemic progression is found to be more rapid than typical, reflecting on the effectiveness of the control measures being taken.

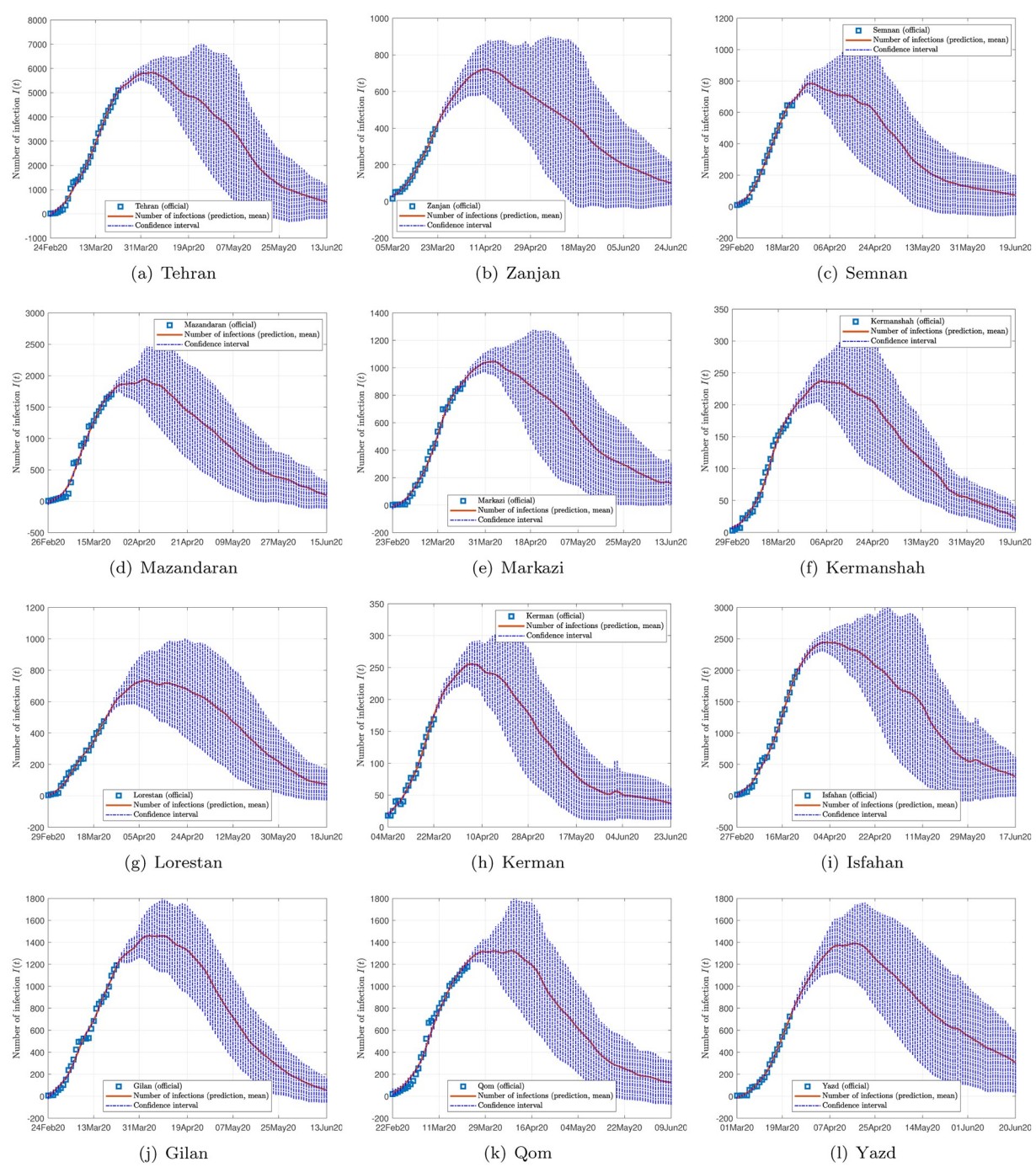

**Fig 3. Official and estimated number of infected individuals in some provinces in Iran.**

Italy has the second highest death toll after China, reaching 197 on March 6, 2020 [19]. The fatality rate is about 4%, which is the highest in the world. With infection cases soaring to 3,916 (on March 6, 2020), Italy had implemented control measures to contain the spread of the virus by shutting down schools and suspending public events in regions where outbreaks

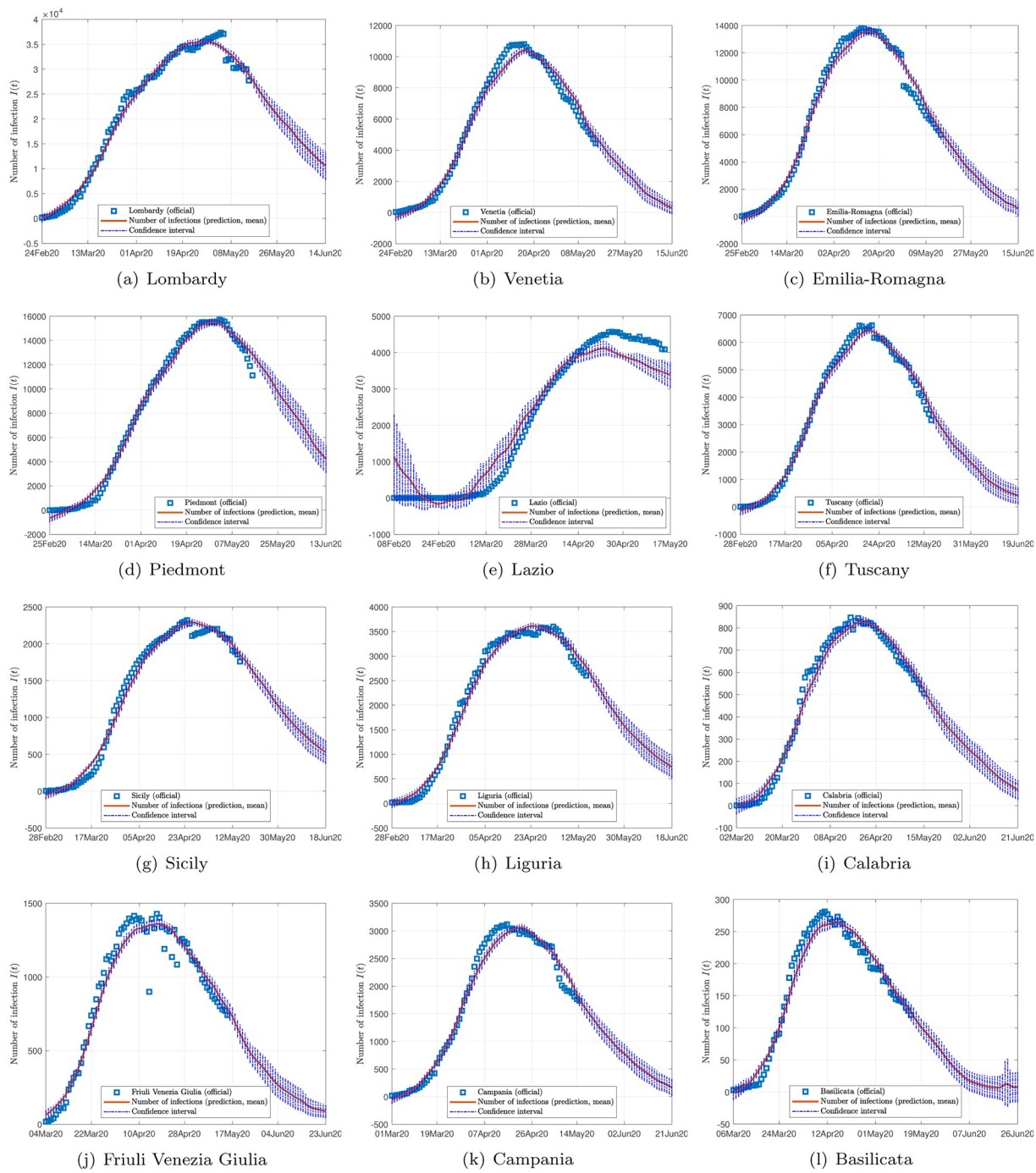

**Fig 4. Official and estimated number of infected individuals in some regions in Italy.**

were reported. The epidemic has been expected to progress in a typical pace (with the present set of parameters), unless more stringent measures are in place.

The situation in Iran is also critical, with the number of infected cases escalated to over 4,000 in less than 2 weeks. Iran has reported death of two lawmakers as of March 7, 2020, and

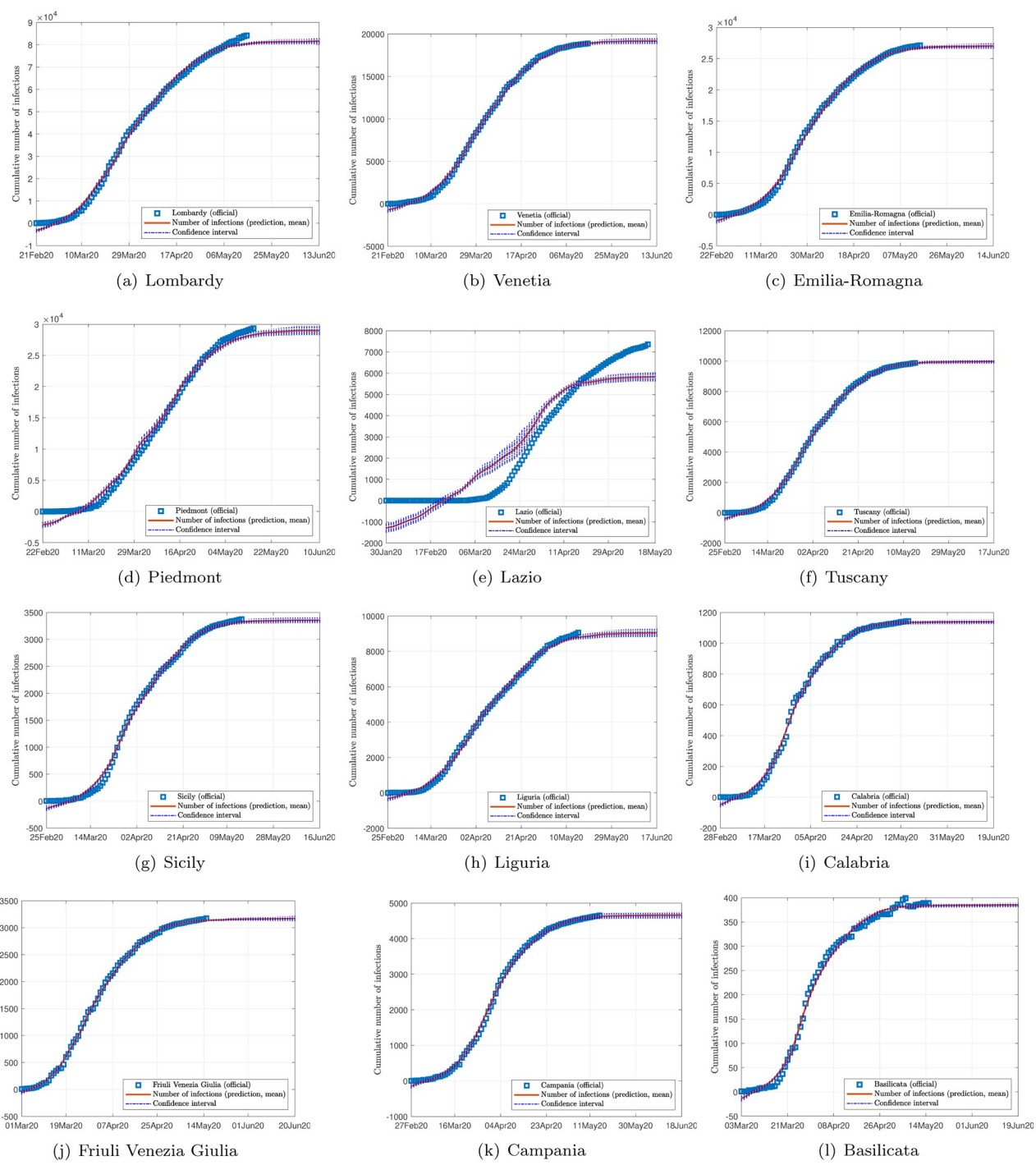

**Fig 5. Official and estimated cumulative number of infected individuals in some regions in Italy.**

has been struggling to control the contagion, which has spread to 31 provinces [20]. The progression profile is again typical, however, expecting to peak in around 3 weeks. Like Italy, most cities show typical spreading profiles, and the peaks and subsequent decline are not expected to advance sooner unless more stringent measures are implemented to control the contagion.

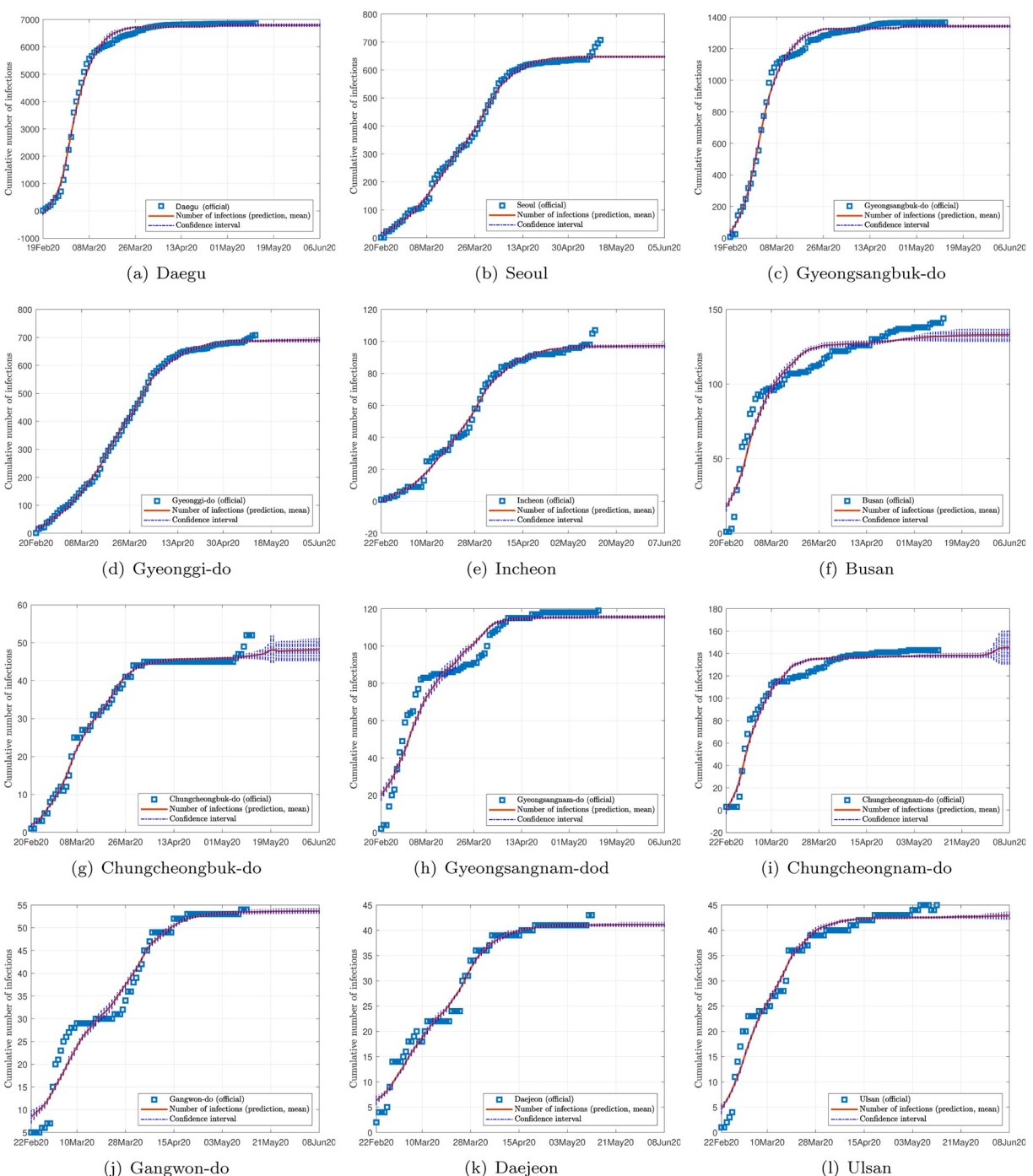

**Fig 6. Official and estimated cumulative number of infected individuals in some cities or regions in South Korea.**

Finally, depending on the effectiveness of treatment, recovery rates vary, and judging from the predicted trends shown in Figs 2, 3 and 4, the epidemic progressions for the three countries are expected to subside by the end of May, with South Korea expected to recover sooner than the others.

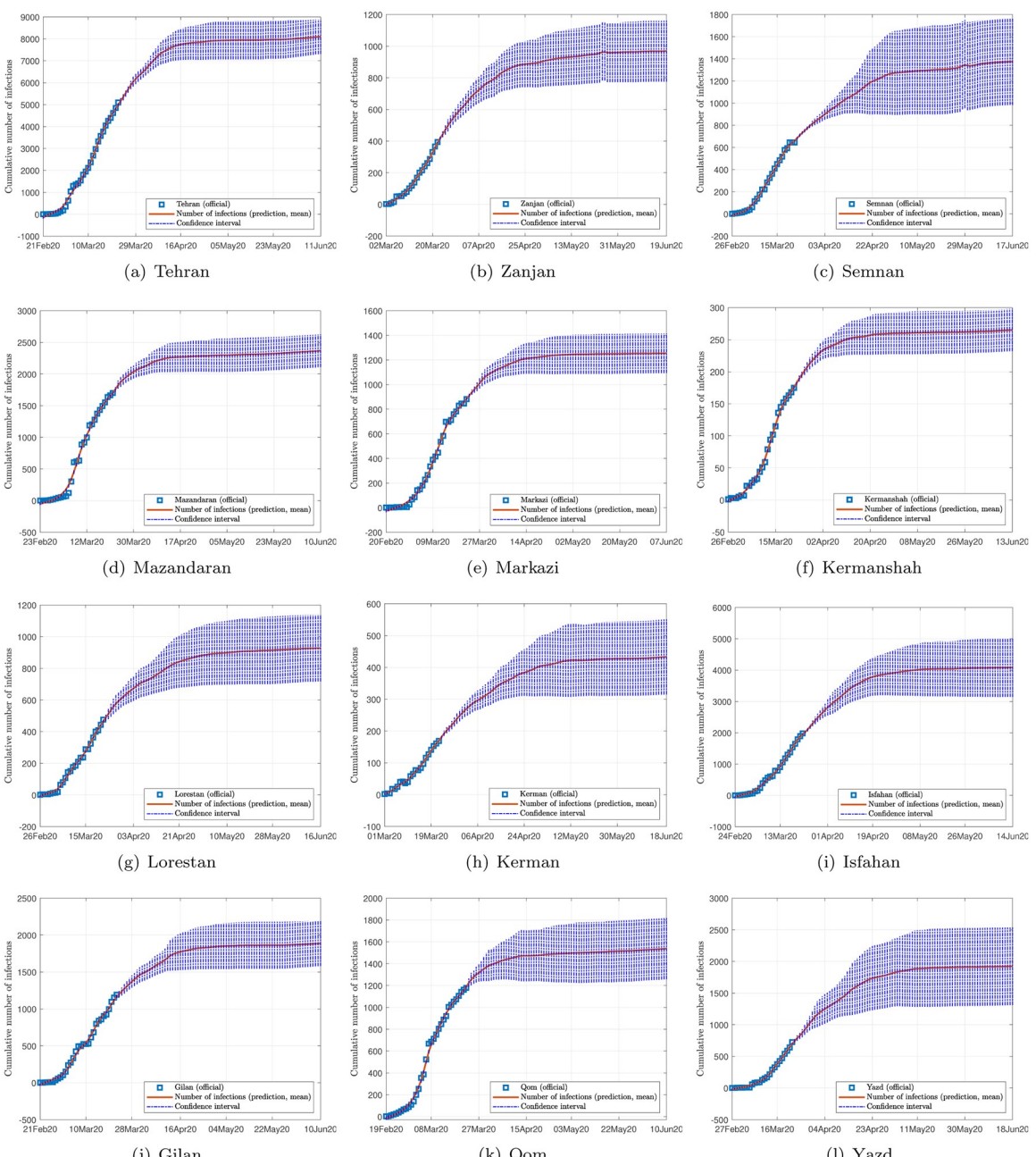

**Fig 7. Official and estimated cumulative number of infected individuals in some provinces in Iran.**

## 5 Conclusion

The spreading of the 2019 New Coronavirus Disease (COVID-19) has evolved to a global contagion, which has spread to 87 countries within two months and more 190 countries until May 14, 2020. The numbers of confirmed infection cases in South Korea, Italy and Iran have surged in late February and early Match, and continued to progress in the last two months, reaching 10,926, 222,104 and 112,725, respectively, on May 14, 2020. The global fatality rate, however,

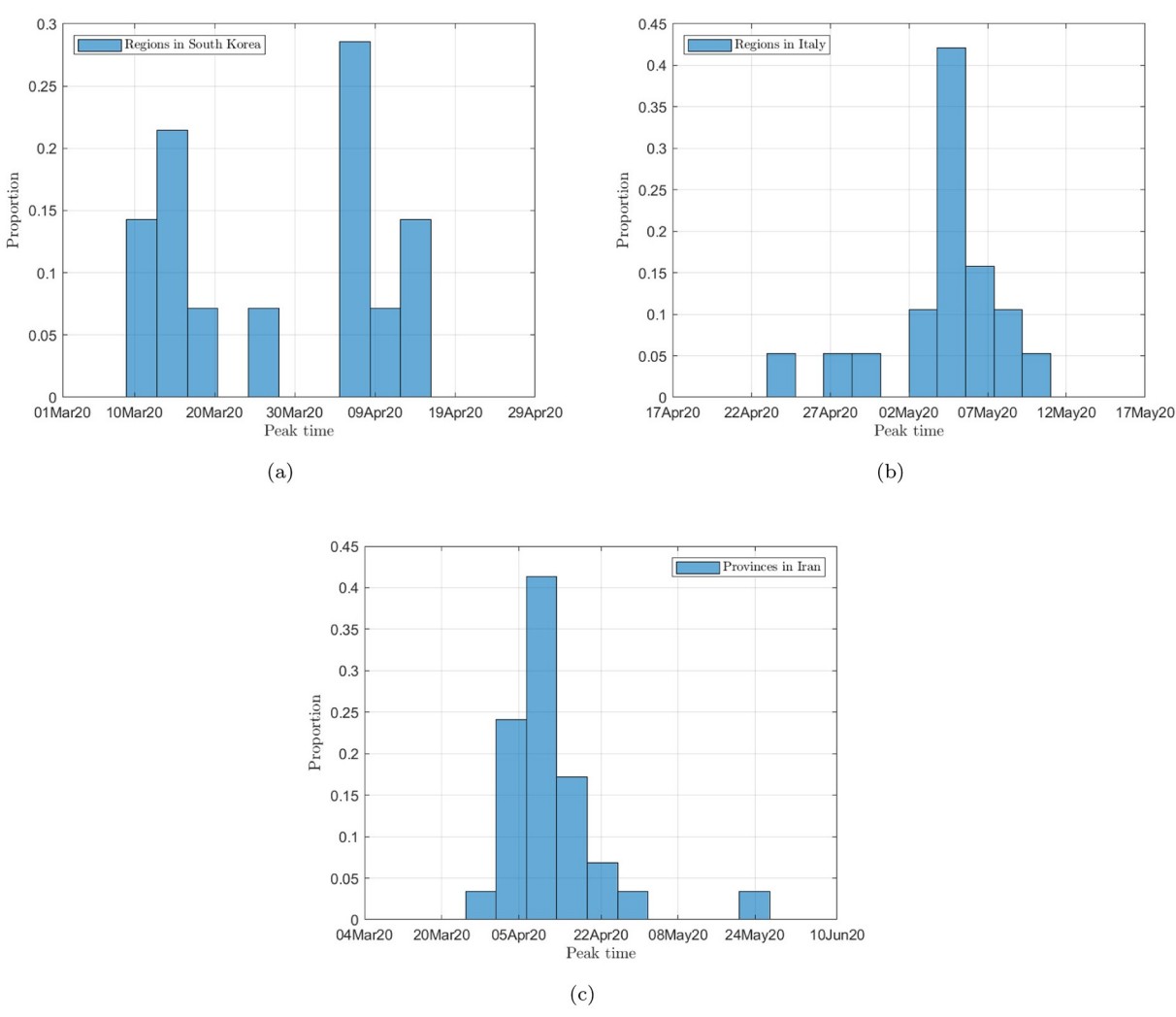

**Fig 8. Statistics of peak times in (a) South Korea; (b) Italy; (c) Iran.**

has increased from below 3% on March 8, 2020 to more than 6% on May 14, 2020. In this study, we build on the results of our previous work [3] that have established a library of parameters of an augmented SEIR model, corresponding to the historic spreading profiles of 367 cities in China. This library forms a set of profile codes that cover a variety of possible epidemic progression profiles. By comparing the early incomplete data of epidemic progression collected for a specific population with the historic profiles, we select a few candidate profiles from the historic archive using a nonlinear optimization procedure. The corresponding profile codes of the selected historic progression profiles can then be used to produce estimates of the future progression for that specific population. We apply this method to predict the spreading profiles for South Korea, Italy and Iran, and specifically, to provide a method for estimating the proportion of infected population in these countries. Results have shown that the three countries would see infection peaks in most cities or provinces before the end of May 2020, with South Korea's cases reaching their peaks much earlier than the others. The percentage of population eventually infected will be less than 0.3%, 0.5% and 0.5% for South Korea, Italy and Iran, respectively. The epidemic is expected to come under control before June 2020 in these

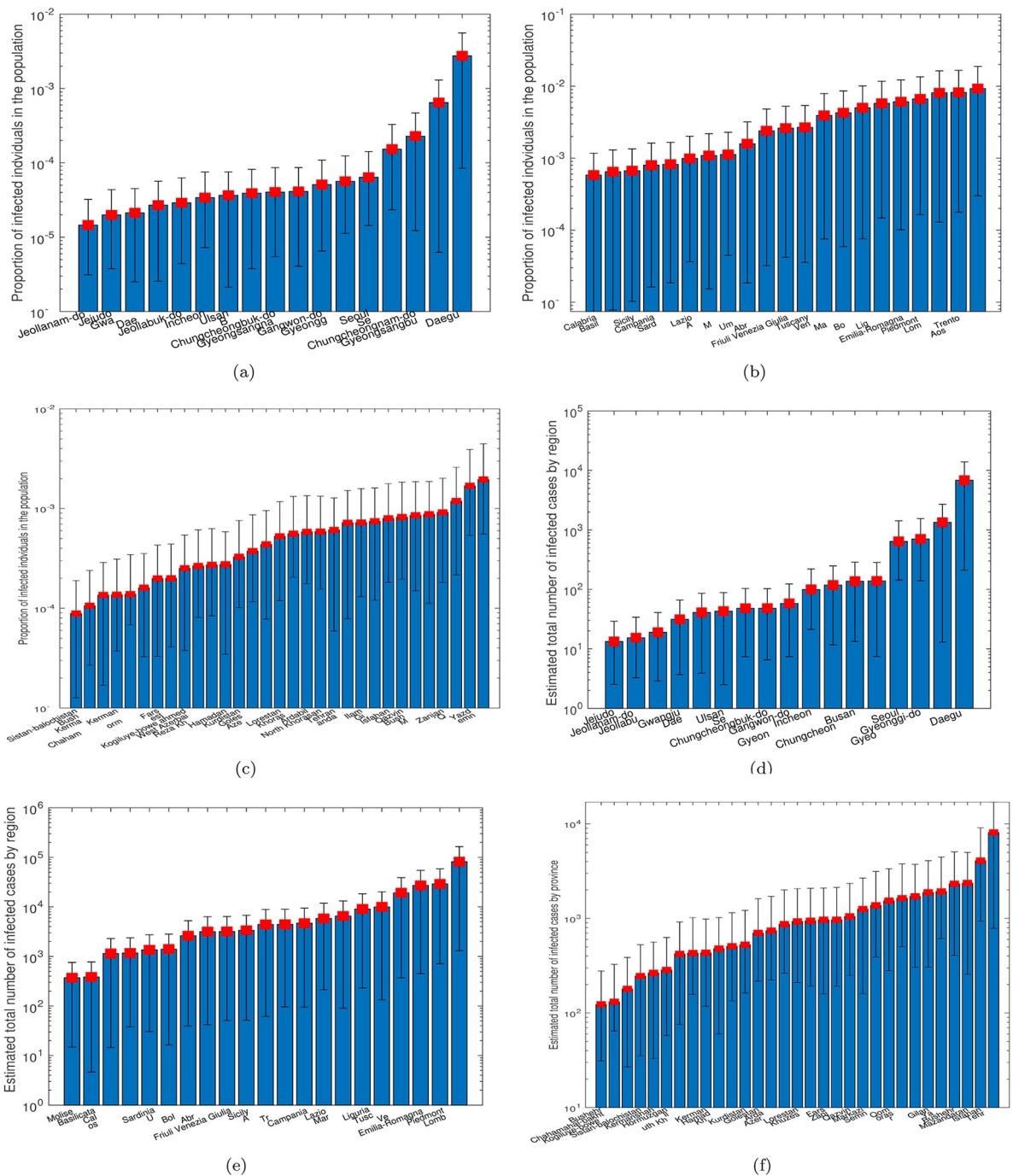

**Fig 9. Statistics of proportion of eventual infected population in (a) South Korea; (b) Italy; (c) Iran. Statistics of eventual infected population in (d) South Korea; (e) Italy; (f) Iran.**

countries, and depending on the effectiveness of treatment, particular cities may see full recovery or zero infection sooner or later than others. It is worth noting that the epidemic progression in South Korean cities are found to be more rapid than typical, implying that the authorities might have taken effective measures to control the spread. The predicted

progressions for Italy and Iran, on the other hand, are found to display profiles that are typical of those in the historical archive, and unless more stringent measures are taken, the peaks and subsequent decline of the infection numbers will unlikely come sooner or more rapidly than the predicted trajectories.

Finally, we should stress that the proposed data-driven coding method is applicable to predicting epidemic progression in any given population and the accuracy of prediction will depend on the adequacy of the available data in allowing a reliable match to be identified from the historical archive. In this study, the profile predicted in early March turned out to be consistent with the actual data collected up to mid May for the three countries. However, due to limited coverage of the data collected, the data-driven model may not perform satisfactorily if it is applied to a new epidemic which has a significantly different set of spreading characteristics (i.e., model parameters being significantly different from those collected in the historical database) or in a population or country having a significantly different contact topology, social behavior, travel patterns, effectiveness of control as well as climate. In fact, deviation was observed in the Hong Kong data in late March when compared with the profile predicted earlier in February, and the deviation was found to be a result of an unexpected surge in inbound travelers which were mostly overseas students returning from the UK and USA. Thus, had superspreader events or other unexpected events occurred, the spreading profile predicted by the data-driven model might deviate from the actual pattern. Furthermore, the model does not consider the effects of different control strategies. Continuous effort will be made to enhance the database so as to widen the coverage of possible progression profiles as well as to incorporate the effects of different control measures on the epidemic progression profiles.

## Supporting information

**S1 File.**
(ZIP)

## Author Contributions

**Conceptualization:** Choujun Zhan, Chi K. Tse.

**Data curation:** Choujun Zhan, Zhikang Lai.

**Formal analysis:** Choujun Zhan, Chi K. Tse.

**Funding acquisition:** Choujun Zhan, Chi K. Tse.

**Investigation:** Choujun Zhan, Chi K. Tse, Zhikang Lai, Tianyong Hao, Jingjing Su.

**Methodology:** Choujun Zhan, Chi K. Tse.

**Project administration:** Choujun Zhan, Chi K. Tse.

**Software:** Choujun Zhan, Zhikang Lai, Tianyong Hao.

**Supervision:** Chi K. Tse.

**Validation:** Choujun Zhan, Zhikang Lai, Tianyong Hao, Jingjing Su.

**Visualization:** Tianyong Hao.

**Writing – original draft:** Chi K. Tse.

**Writing – review & editing:** Choujun Zhan, Chi K. Tse.

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
