## [Decision Letter · Decision Letter 0]

23 Apr 2020

PONE-D-20-06792

Prediction of COVID-19 Spreading Profiles in South Korea, Italy and Iran by Data-Driven Coding

PLOS ONE

Dear prof Chi K. TSE, the manuscript is very interesting.

Please, revise according to the referess points:

1. present more clear the results

2. update the results with today data

3. give a prospetive of your findings

Thank you for submitting your manuscript to PLOS ONE. After careful consideration, we feel that it has merit but does not fully meet PLOS ONE’s publication criteria as it currently stands. Therefore, we invite you to submit a revised version of the manuscript that addresses the points raised during the review process.

We would appreciate receiving your revised manuscript by Jun 07 2020 11:59PM. To enhance the reproducibility of your results, we recommend that if applicable you deposit your laboratory protocols in protocols.io, where a protocol can be assigned its own identifier (DOI) such that it can be cited independently in the future. For instructions see: http://journals.plos.org/plosone/s/submission-guidelines#loc-laboratory-protocols

We look forward to receiving your revised manuscript.

Kind regards,

Delia Goletti, M.D., Ph.D.

Academic Editor

PLOS ONE

Journal Requirements:

Reviewers' comments:

Reviewer's Responses to Questions

**Comments to the Author**

1. Is the manuscript technically sound, and do the data support the conclusions?

Reviewer #1: Partly

Reviewer #2: Partly

2. Has the statistical analysis been performed appropriately and rigorously? 

Reviewer #1: Yes

Reviewer #2: N/A

3. Have the authors made all data underlying the findings in their manuscript fully available?

Reviewer #1: Yes

Reviewer #2: No

4. Is the manuscript presented in an intelligible fashion and written in standard English?

Reviewer #1: Yes

Reviewer #2: Yes

5. Review Comments to the Author

Reviewer #1: In the submitted manuscript, the authors provide a model of prediction of SARS-CoV-2 spreading in three main Countries: South Korea, Italy, and Iran. Their model is based on a previous publication by the same authors and on data collected in 367 Chinese cities. The data and the conclusion are clearly presented and the manuscript is easy to read, but some points have to be clarified and better discussed, especially in the light of updated observed data of viral spreading in the regions under investigation.

One point that can be better explained, to make it easier for the large audience of Plos One to understand the hypothesis of the authors, are the variables taken into account for the construction of the model. Indeed some social/political/climatic variables could differently affect viral spreading in different countries and/or cities. The effect of such factors should be at least discussed in the conclusions.

Another important point to discuss is the difference between the predictions presented and the actual situation (see for example line 200-205). We do understand that the authors performed the analysis having the data till the 6th of March but they now have access to the observed cases in the different countries and regions and some of the observations importantly deviate from the model of spread here presented. The information on observed cases should be updated and discussed proposing criteria for the adjustment of the presented model.

In paragraph line 212-215, why do the authors conclude that the epidemic will end before June 2020, on the basis of which data?

Minor points:

To speak about SARS-CoV-2 spreading would be more appropriated than COVID-19 spreading.

To indicate that for infected individuals the authors mean the people tested positive would also be appropriated, especially when presenting the data as proportion of population infected.

Indeed, only seroprevalence studies may actually estimate the proportion of population that underwent infection.

The names of Italian administrative regions should be corrected in text and figures:

Lombardy, Emilia Romagna, Apulia, Basilicata.

Reviewer #2: The manuscript describes “Prediction of COVID-19 Spreading Profiles in South Korea, Italy and Iran by Data-Driven Coding.” The title is interesting, however, the manuscript is not well organized and understandable because of the lack of a good explanation of the results. The population of each city, province, and region (with a reference) should be mentioned in the results. For example, lines 194-197, the population of Daegu and Seoul should be mentioned. If Seoul’s population is x million with x people eventually infected, it is x% of the city’s population. In addition, it seems that the mentioned places of Iran are Provinces (not cities) and the population of whole province should be considered, however, they should be spelled also correctly (e.g. Fig.6, f, h, i, k, l). Moreover, some parts of the manuscript do not have related references which should be considered. Finally, the limitations of the present study should be addressed.

6. PLOS authors have the option to publish the peer review history of their article (what does this mean?). If published, this will include your full peer review and any attached files.

Reviewer #1: No

Reviewer #2: No

---

## [Author Response · Author response to Decision Letter 0]

19 May 2020

(Already uploaded with the revised manuscript)

We would like to thank the Editor and all reviewers for their helpful comments and suggestions, which have been taken into consideration in the revision of this paper. 

Reviewer: 1

Comments to the Author

In the submitted manuscript, the authors provide a model of prediction of SARS-CoV-2 spreading in three main Countries: South Korea, Italy, and Iran. Their model is based on a previous publication by the same authors and on data collected in 367 Chinese cities. The data and the conclusion are clearly presented and the manuscript is easy to read, but some points have to be clarified and better discussed, especially in the light of updated observed data of viral spreading in the regions under investigation.

Authors’ Response: Thank you for the positive support and encouragement.

One point that can be better explained, to make it easier for the large audience of Plos One to understand the hypothesis of the authors, are the variables taken into account for the construction of the model. Indeed some social/political/climatic variables could differently affect viral spreading in different countries and/or cities. The effect of such factors should be at least discussed in the conclusions.

Authors’ Response: Thank you for the positive support and encouragement. The point raised is indeed very legitimate for data-driven model. We have included a brief discussion at the end of the paper (Conclusion) to highlight this issue, and specifically, performance of any data-driven model would depend on the breath of coverage of the historical data. Thus, we do admit that the model needs to be continuously updated to cover different set of spreading characteristics. At present, we the data is limited, and the data-driven model may not perform satisfactorily if it is applied to a new epidemic which (i.e., model parameters being significantly different from those collected in the historical database) or in a population or country having a significantly different contact topology, travel patterns, effectiveness of government's control as well as climate. 

Another important point to discuss is the difference between the predictions presented and the actual situation (see for example line 200-205). We do understand that the authors performed the analysis having the data till the 6th of March but they now have access to the observed cases in the different countries and regions and some of the observations importantly deviate from the model of spread here presented. The information on observed cases should be updated and discussed proposing criteria for the adjustment of the presented model.

Authors’ Response: The manuscript was finished on March 8, 2020. Now, two months have passed. We have updated the results based on new datasets. All the figures are updated. It is found that the general profile pattern remains the same for the three countries under study, despite some adjustment on the actual number of infected cases predicted and the exact times of the peaks. We believe that the progression profile is basically being captured by the historical data of the 367 cities collected. However, deviation can still be expected due to outlier events such as superspreader events and irregular travel patterns that may cause deviation from the general patterns. For instance, we found significant deviation in the Hong Kong data compared with a similar prediction conducted earlier, and there was unexpected surge in the number of infected cases in mid March due to an unexpectedly large number of inbound travelers as a result of overseas students returning from UK and USA. 

Regarding the updates in this revised version, we should mention that after March 22, Iran no longer publishes detailed data for individual provinces. Thus, we cannot achieve detailed data for individual provinces for forecasting after March 22. The main updates have been included in the blue texts in the Abstract, Section 4 and Conclusion of the revised paper.

In paragraph line 212-215, why do the authors conclude that the epidemic will end before June 2020, on the basis of which data?

Authors’ Response: We provided further explanation in the revised paper as follows. Basically, from the progression trends of the epidemic these three countries, provided control measures continue to be in place, our model show that the number of confirmed cases of COVID-19 infection in most regions of these three countries will peak before the end of May 2020. Hence, the first wave of epidemic progression would come under control before the end of May 2020. This has been mentioned in Page 3 and Page 7 of the revised paper.

Minor points:

To speak about SARS-CoV-2 spreading would be more appropriated than COVID-19 spreading. 

Authors’ Response: According to WHO (https://www.who.int/emergencies/diseases/novel-coronavirus-2019/technical-guidance/naming-the-coronavirus-disease-(covid-2019)-and-the-virus-that-causes-it), the disease is abbreviated as COVID-19 and the virus is SARS-CoV-2. We have found that the literature has more adopted “COVID-19 spreading”. In the revised paper, we have added SARS-CoV-2 in the Introduction but has continued to adopt only COVID-19 for simplicity. 

To indicate that for infected individuals the authors mean the people tested positive would also be appropriated, especially when presenting the data as proportion of population infected. Indeed, only seroprevalence studies may actually estimate the proportion of population that underwent infection.

Authors’ Response: Thank you for pointing this out. In Section 2, as far as our study is concerned, we have clarified the kind of data we have, corresponding to the number of infected cases. See Section 2 on Page 3 of the revised paper. 

The names of Italian administrative regions should be corrected in text and figures:

Lombardy, Emilia Romagna, Apulia, Basilicata.

Authors’ Response: We have tried to unified all the names for the cities and regions of the three countries.

Reviewer: 2

Comments to the Author

The manuscript describes “Prediction of COVID-19 Spreading Profiles in South Korea, Italy and Iran by Data-Driven Coding.” The title is interesting, however, the manuscript is not well organized and understandable because of the lack of a good explanation of the results. 

Authors’ Response: We have updated the results and improved the descriptions in the revised paper, so that the idea of the proposed data-driven model is more clearly explained to the readers. The updates are highlighted in blue in the revised data. We have included more information, as suggested, such as information about populations of individual regions.

The population of each city, province, and region (with a reference) should be mentioned in the results. For example, lines 194-197, the population of Daegu and Seoul should be mentioned. If Seoul’s population is x million with x people eventually infected, it is x% of the city’s population. 

Authors’ Response: See Table 1 of the revised paper. We have also modified the description in the text as per your suggestion.

In addition, it seems that the mentioned places of Iran are Provinces (not cities) and the population of whole province should be considered, however, they should be spelled also correctly (e.g. Fig.6, f, h, i, k, l). 

Authors’ Response: Table 1 indeed are provinces’ populations.

Moreover, some parts of the manuscript do not have related references which should be considered. 

Authors’ Response: The following references have been cited:

• Jia JS, Lu X, Yuan Y, Xu G, Jia J, Christakis NA. Population ow drives spatio-temporal distribution of COVID-19 in China. Nature 2020, 29: 1-1. doi: https://doi.org/10.1038/s41586-020-2284-y10.1126/science.aba9757.

• Du Z, Wang L, Cauchemez S, Xu X, Wang X, Cowling BJ and Meyers LA. Risk of transportation of 2019 novel coronavirus disease from Wuhan to other cities in China. Emerg Infect Dis 2020; 26(5) DOI: 10.3201/eid2601.200146.

• Wu JT, Leung K, Leung GM. Nowcasting and forecasting the potential domestic and international spread of the 2019-nCoV outbreak originating in Wuhan, China: a modelling study. The Lancet 2020; 395(10225): 689-97.

• Humanity tested. Nat Biomed Eng 2020; 4: 355-356.

• Zhan C, Tse C, Lai Z, Chen X and Mo M. General model for COVID-19 spreading with consideration of intercity migration, insuffcient testing and active intervention: application to study of pandemic progression in Japan and USA, Preprint available at medRxiv.org, 2020, doi: https://doi.org/10.1101/2020.03.25.20043380.

Finally, the limitations of the present study should be addressed.

Authors’ Response: Indeed, our data-driven model does have its weakness. due to limited coverage of the data collected, the data-driven model may not perform satisfactorily if it is applied to a new epidemic which has a significantly different set of spreading characteristics (i.e., model parameters being significantly different from those collected in the historical database) or in a population or country having a significantly different contact topology, social behavior, travel patterns, effectiveness of control as well as climate. In fact, deviation was observed in the Hong Kong data in late March when compared with the profile predicted earlier in February, and the deviation was found to be a result of an unexpected surge in inbound travelers which were mostly overseas students returning from the UK and USA. Thus, had superspreader events or other unexpected events occurred, the spreading profile predicted by the data-driven model might deviate from the actual pattern. We have provided some comments in the Conclusion.

---

## [Editor Report · Decision Letter 1]

3 Jun 2020

Prediction of COVID-19 Spreading Profiles in South Korea, Italy and Iran by Data-Driven Coding

PONE-D-20-06792R1

Dear Dr. Chi K. TSE, 

We’re pleased to inform you that your manuscript has been judged scientifically suitable for publication and will be formally accepted for publication once it meets all outstanding technical requirements.

Kind regards,

Delia Goletti, M.D., Ph.D.

Academic Editor

PLOS ONE
---

## [Editor Report · Acceptance letter]

25 Jun 2020

PONE-D-20-06792R1 

Prediction of COVID-19 Spreading Profiles in South Korea, Italy and Iran by Data-Driven Coding 

Dear Dr. Tse:

I'm pleased to inform you that your manuscript has been deemed suitable for publication in PLOS ONE. Congratulations! Your manuscript is now with our production department. 

Kind regards, 

on behalf of

Dr. Delia Goletti 

Academic Editor

PLOS ONE